# OpenReview forum: "SOPBench: Evaluating Language Agents at Following Standard Operating Procedures and Constraints"
_ICLR.cc/2026/Conference — Submitted to ICLR 2026_

### Official Review · Reviewer_o9er · 2025-10-26

**Soundness:** 3
**Presentation:** 3
**Contribution:** 2
**Rating:** 4
**Confidence:** 4

**Summary:**

The paper introduces SOPBench, a benchmark designed to evaluate the ability of language agents to follow standard operating procedures (SOPs) and constraints in customer service domains. It addresses the growing need for agents to adhere to domain-specific procedural guidelines, a capability underexplored in existing evaluations. The approach includes sandbox environments with 167 executable tools across seven domains, an automated test generation framework creating over 800 test cases, and a multi-level evaluation harness using oracle code as verifiers. The study finds that even top-tier models like o4-mini-high struggle, with pass rates around 30% in challenging domains, and highlights vulnerabilities to jailbreaking, emphasizing the need for improved adherence in high-stakes environments.

**Strengths:**

1. The development of a novel evaluation pipeline using executable SOP code as oracle verifiers enhances the reliability and objectivity of assessing agent compliance, reducing dependence on subjective human or LLM-based judgments. This approach ensures consistent and scalable evaluation across diverse scenarios, making it a robust tool for future research.

2. The inclusion of 167 executable tools and 70 service-specific SOPs across seven customer service domains provides a comprehensive and realistic testing ground, covering areas like banking and healthcare. This breadth allows for a thorough assessment of agent capabilities, revealing domain-specific challenges and opportunities for improvement.

3. The multi-level verification process, including outcome, step, and trajectory-level checks, offers a detailed analysis of agent adherence to SOPs, capturing both final results and procedural accuracy. This granularity provides valuable insights into agent decision-making, aiding in the identification of specific weaknesses.

4. The release of code, data, and over 24k agent trajectories fosters transparency and enables the research community to replicate and build upon the findings. This open-access strategy promotes collaboration and accelerates advancements in language agent development.

**Weaknesses:**

1. The reliance on manually implemented sandbox environments and SOPs across seven domains may introduce potential biases or inconsistencies, as human design could overlook certain real-world complexities. This limitation might affect the generalizability of the results to less structured or more dynamic settings.

2. The benchmark's focus on customer service domains, while comprehensive, may not fully represent the diversity of tasks in other critical areas like healthcare diagnostics or legal processes, potentially limiting its applicability. This narrow scope could miss unique procedural challenges in those fields.

3. The finding that top-tier models achieve pass rates below 70% in challenging domains indicates that the benchmark's difficulty may exceed current model capabilities, potentially discouraging progress rather than guiding it. This high difficulty could lead to frustration rather than constructive development.

4. The lack of detailed analysis on the impact of different tool-calling methods (e.g., ReAct vs. Act-Only) beyond basic comparisons limits insights into optimizing agent performance, leaving a gap in understanding how methodological choices affect SOP adherence. This omission could hinder tailored improvements for specific models.

**Questions:**

See weaknesses.

---

> ### Author Response · Authors · 2025-11-28
> **Response to reviewer o9er [1/2]**
>
> We thank the reviewer for the detailed summary and for recognizing the strengths of SOPBench, including the reliability of executable oracle verifiers, the breadth of domains and tools, the multi-level evaluation design, and the open release of datasets and trajectories. We appreciate these positive remarks and address the reviewer’s concerns below.
>
> > Re C1: Potential biases or limitations from manually implemented sandbox environments and SOPs.
>
> We agree that manually implemented environments may miss certain real-world nuances. However, our primary goal in SOPBench is to construct deterministic, executable, and verifiable SOPs that support unambiguous, code-based evaluation—something difficult to guarantee with organically collected SOPs. This controlled setup enables precise assessment of a model’s ability to follow SOPs and constraints when using tools to complete tasks. Each domain underwent multiple rounds of iteration and manual review to minimize overlooked cases.
>
> > Re C2: Focus on customer service domains may limit broader applicability.
>
> Customer-service settings were chosen because they provide rich multi-step SOPs, diverse constraint types, and realistic task structures, making them well suited for an initial benchmark. However, SOPBench is designed as an extensible framework: adding a new domain only requires specifying the constraint graph, service/helper functions, and database schema. Our pipeline then automatically generates the corresponding test cases (see Section 2.4, Figure 4, and details in Appendix B). With the code fully open-sourced, both we and the community can readily incorporate additional domains such as legal workflows, healthcare diagnostics, HR processes, or IT operations.
>
> > Re C3: High benchmark difficulty may exceed current model capabilities.
>
> Our benchmark uniquely reveals that current models and efforts focus primarily on using tools to complete tasks, while largely overlooking the need to follow SOPs and constraints to complete tasks safely and permissibly. We therefore view the benchmark’s difficulty not as a barrier, but as an important diagnostic signal highlighting this gap.
>
> Moreover, our evaluation across 18 models shows clear separability in performance, revealing meaningful differences in their procedural-reasoning and SOP-following capabilities. This indicates substantial headroom for future improvement—rather than all models performing uniformly poorly, which would indeed suggest an unproductive barrier. We believe that SOPBench thus provides actionable insight rather than an insurmountable challenge.
>
> > Re C4: Limited analysis of different tool-calling methods (ReAct vs. Act-Only).
>
> Current models are generally optimized for Function Calling (FC) formats, so it is expected that FC outperforms ReAct and Act-Only. Nonetheless, the comparison between ReAct and Act-Only, and the stronger results from reasoning models such as o4-mini and DeepSeek-R1, underscores the importance of reasoning and planning in SOPBench.
>
> Sepcfially, task difficulty in SOPBench is multifaceted, as each task requires the model to both *remember* to follow the SOP and *correctly* execute it. Concretely, models must demonstrate three core capabilities:
>  - **C1: Complete constraint verification**: remember to check *all* relevant constraints.
>  - **C2: Correct constraint reasoning**: select the appropriate helper function for each constraint and correctly interpret its output to determine whether the constraint is satisfied, and finally determine whether the target service action is permissible.
>  - **C3: Safe and compliant action execution**: execute only permissible user requests and refuse impermissible ones, balancing helpfulness with safety.
>
> To analyze where models fail, we categorize errors into five types, each aligned with the capabilities above:
>  - **Completely overlook constraints**: the model ignores all constraints and directly calls the target service function (C1).
>  - **Partially overlook constraints**: the model attempts constraint verification but fails to check all required constraints (C1).
>  - **Wrong interpretation**: the model calls the correct helper functions but misinterprets their outputs, leading to an incorrect final decision (C2).
>  - **Fail to complete a permissible task**: the model should execute the user request but fails to do so (C3).
>  - **Complete an impermissible task**: the model executes the target service function even though constraints prohibit it (C3).
>
> [1/2]

---

> > ### Author Response · Authors · 2025-11-28
> > **Response to reviewer o9er [2/2]**
> >
> > Below is the error distribution across representative models:
> >
> > | Model | o4-mini | gemini-2.5-flash | gpt-4.1 | claude-3.7-sonnet | llama3.1-70b-instruct | qwen2.5-32b-instruct | qwen2.5-14b-instruct |
> > | :---- | :---- | :---- | :---- | :---- | :---- | :---- | :---- |
> > | Completely overlook constraints | 50 | 52 | 8 | 30 | 131 | 122 | 95 |
> > | Overlook partial constraints | 22 | 51 | 47 | 81 | 112 | 126 | 161 |
> > | Wrong interpretation | 14 | 22 | 23 | 26 | 31 | 20 | 20 |
> > | Fail to complete permissible task | 87 | 85 | 68 | 65 | 88 | 76 | 66 |
> > | Complete impermissible task | 52 | 90 | 139 | 205 | 219 | 226 | 276 |
> >
> > These models show different error patterns:
> >  - **o4-mini:** Often either checks all constraints or ignores them entirely. Rarely misinterprets results, but is **overly conservative**, failing many permissible tasks and thus reducing helpfulness.
> >  - **Gemini-2.5-Flash:** Misses more constraints than o4-mini and shows a **balanced mix** of failing permissible tasks and completing impermissible ones, indicating unstable safety–helpfulness behavior.
> >  - **GPT-4.1 vs. Claude-3.7-Sonnet:** GPT-4.1 almost never overlooks all constraints, though it still misses partial checks. Claude-3.7 shows **more overlook errors** overall, reflecting weaker constraint-following ability. Both models more frequently **complete impermissible tasks**, suggesting safety lapses after partial reasoning.
> >  - **Open-source models (Llama3.1-70B, Qwen2.5-32B/14B):** Frequently overlook constraints (partially or fully) and have the **highest impermissible completion rates**, indicating weak structural understanding of constraints and poor safety compliance.
> >
> > In summary, while prompting formats do influence behavior, the dominant factor in SOPBench performance might be a model's internal ability to conduct multi-step SOP reasoning and adherence. Our analysis here aims to highlight these underlying capability gaps.

---

### Official Review · Reviewer_8YNX · 2025-10-28

**Soundness:** 3
**Presentation:** 3
**Contribution:** 2
**Rating:** 4
**Confidence:** 3

**Summary:**

This paper introduces SOPBench, an automated evaluation pipeline designed to assess language agents' ability to follow domain-specific standard operating procedures (SOPs), policies, and constraints. SOPBench features executable environments with 167 tools/functions across seven customer service domains, an automated test generation framework with over 900 verified test cases, and a rigorous evaluation framework. The approach transforms SOP code into directed graphs of executable functions, requiring agents to act based on natural language SOP descriptions. The original code serves as oracle rule-based verifiers, reducing reliance on manual annotation. The authors evaluate 18 leading models, finding that even top-tier models struggle with SOP adherence, and that agents are susceptible to jailbreaking. The dataset and code are released for further research.

**Strengths:**

Originality: The focus on SOP adherence is novel and underexplored in existing benchmarks.
Quality: The automated, rule-based evaluation framework is robust and scalable.
Clarity: The methodology and results are clearly presented.
Relevance: The benchmark and findings are highly relevant for the development and deployment of reliable language agents.

**Weaknesses:**

The paper could provide more analysis on why certain models fail specific SOPs and constraints.

Limited discussion on the generalizability of SOPBench to domains beyond customer service.

The susceptibility to jailbreaking is noted, but mitigation strategies are not explored in depth.

Some experimental details (e.g., SOP selection criteria, test case diversity) could be expanded.

**Questions:**

My major question is if a task is governed by a well-defined SOP or protocol, is it necessary—or even optimal—to use a Large Language Model (LLM) to perform it, rather than simply encoding the SOP directly in code? Take the minor declaration task shown in Fig. 2 as an example, would the task be solved more easily by creating a functionality in the university admin website which can let the students just select the minor they want to declare, and do all those checks in the backend database? IMHO, creating this functionality by coding with the assistance of LLM could be more efficient and straightforward than going through the LLM agent.

My other minor questions are
Can you elaborate on the criteria used to select SOPs and domains for inclusion in SOPBench?

How does SOPBench handle ambiguous or conflicting SOPs in real-world scenarios?

How does SOPBench compare to existing agent evaluation benchmarks in terms of coverage and difficulty?

---

> ### Author Response · Authors · 2025-11-28
> **Response to reviewer 8YNX [1/3]**
>
> We thank the reviewer for the thoughtful summary and for highlighting the strengths of SOPBench, including its novelty, robust rule-based evaluation, clear methodology, and relevance for developing reliable language agents. We appreciate these encouraging remarks and address the reviewer’s concerns and questions below.
>
> > Re C1: More analysis is needed on why certain models fail specific SOPs and constraints.
>
> Task difficulty in SOPBench is multifaceted, as each task requires the model to both *remember* to follow the SOP and *correctly* execute it. Concretely, models must demonstrate three core capabilities:
>  - **C1: Complete constraint verification**: remember to check *all* relevant constraints.
>  - **C2: Correct constraint reasoning**: select the appropriate helper function for each constraint and correctly interpret its output to determine whether the constraint is satisfied, and finally determine whether the target service action is permissible.
>  - **C3: Safe and compliant action execution**: execute only permissible user requests and refuse impermissible ones, balancing helpfulness with safety.
>
> To analyze where and why models fail, we categorize errors into five types, each aligned with the capabilities above:
>  - **Completely overlook constraints**: the model ignores all constraints and directly calls the target service function (C1).
>  - **Partially overlook constraints**: the model attempts constraint verification but fails to check all required constraints (C1).
>  - **Wrong interpretation**: the model calls the correct helper functions but misinterprets their outputs, leading to an incorrect final decision (C2).
>  - **Fail to complete a permissible task**: the model should execute the user request but fails to do so (C3).
>  - **Complete an impermissible task**: the model executes the target service function even though constraints prohibit it (C3).
>
> Below is the error distribution across representative models:
>
> | Model | o4-mini | gemini-2.5-flash | gpt-4.1 | claude-3.7-sonnet | llama3.1-70b-instruct | qwen2.5-32b-instruct | qwen2.5-14b-instruct |
> | :---- | :---- | :---- | :---- | :---- | :---- | :---- | :---- |
> | Completely overlook constraints | 50 | 52 | 8 | 30 | 131 | 122 | 95 |
> | Overlook partial constraints | 22 | 51 | 47 | 81 | 112 | 126 | 161 |
> | Wrong interpretation | 14 | 22 | 23 | 26 | 31 | 20 | 20 |
> | Fail to complete permissible task | 87 | 85 | 68 | 65 | 88 | 76 | 66 |
> | Complete impermissible task | 52 | 90 | 139 | 205 | 219 | 226 | 276 |
>
> These models show different error patterns:
>  - **o4-mini:** Often either checks all constraints or ignores them entirely. Rarely misinterprets results, but is **overly conservative**, failing many permissible tasks and thus reducing helpfulness.
>  - **Gemini-2.5-Flash:** Misses more constraints than o4-mini and shows a **balanced mix** of failing permissible tasks and completing impermissible ones, indicating unstable safety–helpfulness behavior.
>  - **GPT-4.1 vs. Claude-3.7-Sonnet:** GPT-4.1 almost never overlooks all constraints, though it still misses partial checks. Claude-3.7 shows **more overlook errors** overall, reflecting weaker constraint-following ability. Both models more frequently **complete impermissible tasks**, suggesting safety lapses after partial reasoning.
>  - **Open-source models (Llama3.1-70B, Qwen2.5-32B/14B):** Frequently overlook constraints (partially or fully) and have the **highest impermissible completion rates**, indicating weak structural understanding of constraints and poor safety compliance.
>
> Overall, stronger models tend to remember to check constraints but may misinterpret results or act unsafely afterward. Weaker models often miss constraint checks entirely. These patterns highlight that SOPBench exposes *distinct procedural-reasoning weaknesses* across models, not just aggregate accuracy differences, and pinpoints where future model improvements are needed.
>
> **Regarding constraint difficulty**, we find that temporal constraints, particularly in the Hotel domain, are consistently challenging for all models, whereas other constraint types show more model-dependent variability.
>
> The Hotel domain contains 10 temporal constraints, which span a diverse set of temporal operations, including lead-time windows, hour-level deadlines, intra-day service periods, maximum-stay limits, interval-length calculations, and overlap detection across existing bookings.
>
> [1/3]

---

> > ### Author Response · Authors · 2025-11-28
> > **Response to reviewer 8YNX [2/3]**
> >
> > For example, the `modify_reservation` action requires satisfying four simultaneous temporal constraints:
> >  - Within the lead-time window. The new check-in date must fall within an allowed window relative to “today.” In other words, the guest cannot move the reservation to a date that is too soon or too far in the future.
> >  - No overlap with exclusion. When modifying a reservation, the assistant must ignore the original booking interval and check whether the guest has some other booking whose date range overlaps with the proposed new check-in and check-out dates.
> >  - Before an hour-level deadline. The modification request must be made sufficiently in advance of arrival, measured in hours rather than days.
> >  - Within the maximum stay length. The modified reservation must not exceed the maximum allowed number of nights.
> >
> > > Re C2: Limited discussion on the generalizability of SOPBench to domains beyond customer service.
> >
> > SOPBench is designed as an extendable framework: adding a new domain only requires specifying the constraint graph, service/helper functions, and database schema. Our pipeline then automatically generates the corresponding test cases (see Section 2.4, Figure 4, and details in Appendix B).
> > This structure makes it straightforward to introduce richer domains beyond customer services. With the code fully open-sourced, both we and the community can readily incorporate additional domains.
> >
> >
> > > Re C3: The susceptibility to jailbreaking is noted, but mitigation strategies are not explored in depth.
> >
> > Existing tool-calling leaderboards focus primarily on task completion, whereas SOPBench evaluates whether a model remembers to follow the required SOPs and constraints and can consistently adhere to them, rather than simply fulfilling any user request regardless of safety or permissibility.
> >
> > Our results show that models already struggle with SOP compliance even under normal conditions, without any adversarial intent. Therefore, we see SOPBench as an important first step toward diagnosing and understanding these vulnerabilities; developing and evaluating defense mechanisms is a natural direction for future work. Additionally, exploring mitigation strategies in depth would significantly expand the scope and dilute the focus of this paper, especially under the page limit.
> >
> >
> > > R4 C4: Why use LLM agents instead of directly encoding SOPs in code?
> >
> > We agree that, when a system designer controls the environment (e.g., a university admin website), directly implementing the SOP in backend code is indeed the optimal solution. Our goal, however, as discussed in Line 45 to Line 49, is to evaluate LLM agents operating in open natural-language environments, where backend access is not available and SOPs must be followed based solely on descriptions provided in text.
> >
> > Many real-world scenarios—customer support, workflow delegation, enterprise automation, healthcare triage, IT operations—require agents to interpret and execute SOPs written in natural language, interact with external APIs, and generalize to new or evolving policies without system-level updates. SOPBench is designed to test whether LLM agents can safely and reliably follow such procedural instructions, not to replace traditional coded workflows.
> > Our benchmark provide a controllebd environment where whether they follow the natural-language constraints/SOPs, with the equivement coded workflow as the oracle, to provide real-time, accurate, determintisc evaluation.
> >
> >
> > > Re C5: Some experimental details (e.g., SOP selection criteria, test case diversity) could be expanded.
> >
> > We construct 7 domains—Bank, DMV, Healthcare Insurance, Library, Online Market, Hotel, and University—chosen for their prevalence and relevance to real-world LLM agent deployments. SOPs were selected to represent common operations in these domains and to allow executable verification.
> >
> > As for the test case diversity, as briefly explained in the core paper and expanded further in the appendix, the task selection and diversification is done in the task generation step of our pipeline, which consists of three stages:
> >  - Task Permutation: delegation of the action constraint set and outcome that define the task
> >  - LLM Data Generation: data generation to satisfy the task specifications, including user query, initial DB, etc.
> >  - Data Verification: verifying the LLM generated data satisfies the task specifications by confirming the generated data will lead to the previously enumerated outcomes
> >
> > We ensure task diversity by permutating through possible combinations of applicable constraints as well as the possible constraint outcomes. Each’s task’s constraint dependency or state transition is different.
> >
> > Additionally, we present the graph edit distances (GED) between pairs of all action graphs for each domain:
> > | Domain | GED Median | GED Max |
> > | --- | --- | --- |
> > Bank | 12| 24 |
> > DMV | 5 | 18 |
> > Healthcare | 13 | 37 |
> > Hotel | 8 | 29 |
> > Library | 13 | 38 |
> > Online Market | 20 | 49 |
> > University | 8 | 13 |
> >
> > [2/3]

---

> > > ### Author Response · Authors · 2025-11-28
> > > **Response to reviewer 8YNX [3/3]**
> > >
> > > > Re C6: How does SOPBench handle ambiguous or conflicting SOPs in real-world scenarios?
> > >
> > > Thank you for the suggestion. In the current version of SOPBench, we ensure through manual review that all SOPs are clear, unambiguous, and internally consistent so that deterministic oracle verification is possible.
> > >
> > > We agree that ambiguous and conflicting SOP scenarios would be valuable extensions. We plan to incorporate both in future versions:
> > >  - Ambiguous SOPs: We plan to check whether models hallucinate necessary missing information to complete the tasks or correctly request clarification from the user before proceeding.
> > >  - Conflicting SOPs: We plan to introduce explicit priority rules or conditions to test whether models can identify and apply the correct precedence when SOPs disagree.
> > >
> > >
> > >
> > > > Re C7: How does SOPBench compare to existing agent evaluation benchmarks in terms of coverage and difficulty?
> > >
> > > Existing tool-calling benchmarks such as BFCL-v3 [1], TauBench [2], and Tau2Bench [3] focus primarily on task completion. In contrast, SOPBench evaluates whether a model can remember and consistently adhere to SOPs and constraints, rather than executing any user request regardless of safety or permissibility. Beyond this conceptual difference, we compare coverage and difficulty below.
> > >
> > > **Coverage**:
> > > | Benchmark | Domains | # Tools |
> > > | --- | --- | --- |
> > > | BFCL-v3 [1] | 8 | 84 |
> > > | TauBench [2] | 2 | 28 |
> > > | Tau2Bench [3] | 3 | 68 |
> > > | SOPBench (Ours) | 7 | **167** |
> > >
> > > SOPBench provides far more tools and covers more domains than TauBench/Tau2Bench, with domain coverage comparable to BFCL-v3.
> > >
> > > **Difficulty**:
> > > We compare the performance of the four models evaluated in [3] across BFCL, Tau2Bench (all 3 domains), and SOPBench (full benchmark and Healthcare domain subset), along with SOPBench-adv (with ASAP attack).
> > >
> > > | Model | BFCL | Tau2Bench (all 3 domains) | SOPBench (all 7 domains) | SOPBench (Healthcare) | SOPBench-adv (Healthcare) |
> > > | --- | --- | --- | --- | --- | --- |
> > > | GPT-4.1 | 54.57 | 54.48 | 65.54 | 79.03 | 34.92 |
> > > | o4-mini | 55.57 | 56.99 | 75.52 | 92.74 | 41.27 |
> > > | GPT-4.1-mini | 50.07 | 54.12 | 45.34 | 66.13 | 22.78 |
> > > | Claude-3.7-Sonnet | - | 61.54 | 54.46 | 70.97 | 12.70
> > >
> > > From these results:
> > > (1) BFCL and Tau2Bench offer limited model separability, with models clustering closely together. In contrast, SOPBench produces clearly differentiated performance, revealing substantial capability differences.
> > > (2) Under the simple ASAP attack, all models degrade sharply on SOPBench, and notably, Claude-3.7, strongest on Tau2Bench, performs the worst.
> > >
> > > **These observations suggest that prior efforts have focused primarily on improving tool-use proficiency for task completion, while largely overlooking safe and procedurally correct task execution.** SOPBench directly targets this gap by evaluating adherence to SOPs, constraints, and safety requirements.
> > >
> > >
> > >
> > >
> > > ## References
> > >
> > > [1] Patil, Shishir G., et al. "The Berkeley Function Calling Leaderboard (BFCL): From Tool Use to Agentic Evaluation of Large Language Models." Forty-second International Conference on Machine Learning.
> > >
> > > [2] Yao, Shunyu, et al. "tau-bench: A Benchmark for Tool-Agent-User Interaction in Real-World Domains." arXiv preprint arXiv:2406.12045 (2024).
> > >
> > > [3] Barres, Victor, et al. "tau2-Bench: Evaluating Conversational Agents in a Dual-Control Environment." arXiv preprint arXiv:2506.07982 (2025).

---

### Official Review · Reviewer_h1ei · 2025-11-01

**Soundness:** 3
**Presentation:** 2
**Contribution:** 2
**Rating:** 4
**Confidence:** 4

**Summary:**

The paper presents SOPBENCH, a benchmark for evaluating LLM agents on following.standard operating procedures and constraints. The benchmark is created by first manually defining an environment sandbox for seven domains. The sandbox consists of a set of constraints, 70 service functions and 97 helper functions that can be used to complete the tasks and verify constraints, a database schema containing the necessary information for constraint verification. Based on the environment sandbox, 830 test cases are generated using LLMs and verified using rule-based verifiers. Evaluation is conducted based on three criteria that can be measured using oracle code programs, namely outcome-level verification that verifies the final database state, step-level verification that verifies the constraint permissibility of each function called by the agent, and trajectory-level verification that verifies whether all the verification steps are completed in the correct order. Experimental results show that even SOTA models such as o4-mini-high struggle particularly on more challenging domains like hotel.

**Strengths:**

- Benchmark tackles an important problem of SOP following, which is required for agents to be useful in the real-world
- Scalable and reproducible methodology for automatically generating and validating test cases
- Comprehensive and reliable evaluation using oracle code programs
- Experimental results demonstrate the benchmark complexity and room for future research
- Performance degradation with a simple jailbreaking approach highlights the brittleness of modern LLMs on this task

**Weaknesses:**

- At its core, the task seems similar to structured tool calling. It is unclear as to what differentiates this benchmark from tool calling benchmarks, such as Berkley Function Calling Leaderboard that also conduct state-based evaluation and check whether the model’s function call trajectory includes the minimum necessary path. So, the core novelty of this benchmark, at least in the current scope of composition types, is limited.
- There is some inconsistency between Figures 7 and 11. For example, "show_available_rooms" has almost 0 success rate as per Figure 7 but Figure 11 shows high success for this service task.
- Limited error analysis. While the paper shows the model performance breakdown by evaluation criteria in Figure 8, and one example erroneous trajectory in the Appendix, it is still unclear why leading models fail. Also, what makes domains like Hotel more challenging?

**Questions:**

- How is evaluation done with temp = 0 when models like GPT-5 do not support temperature?
- Since the constraints can be translated into a code program, how is the performance if the LLM is asked to generate code for constraint verification?

---

> ### Author Response · Authors · 2025-11-28
> **Response to reviewer h1ei [1/2]**
>
> We thank the reviewer for the constructive feedback and the recognition of our work’s strengths, including its focus on real-world SOP adherence, scalable methodology, reliable oracle-based evaluation, and demonstrated task complexity. We also appreciate the reviewer noting the brittleness of modern LLMs under simple jailbreak prompts, which further highlights the safety relevance of SOPBench.
>
> Addressing Concerns:
>
> > Re C1: Differences and novelty beyond existing tool-calling benchmarks
>
> Although SOPBench uses function calls as the execution interface, its core challenge is multi-step SOP compliance under constraints in tool use, instead of tool invocation and task completion. **SOPBench evaluates whether a model can safely complete a task by following all required procedures，not merely whether it can use tools to complete whatever the user requests regardless of safety or permissibility.** This focus is crucial for the safe deployment of language agents, as also noted by other reviewers.
>
> Specifically, SOPBench requires models to: (1) identify and verify all relevant constraints in the correct order; (2) call the appropriate helper functions for each constraint check; (3) interpret intermediate results for each check and update decisions accordingly.
>
> Our results show that current strong models excel at tool-use proficiency but often fail at procedurally correct and safety-aware execution. The jailbreak experiment further demonstrates how easily models can be prompted to bypass constraints and SOPs. Together, these findings highlight the critical need for benchmarks like SOPBench that measure, and drive progress on—safe, procedure-following behavior.
>
> > Re C2: "There is some inconsistency between Figures 7 and 11. For example, show_available_rooms has almost 0 success rate in Figure 7 but Figure 11 shows high success."
>
> There are two distinct tasks both named “show_available_rooms”, one in the Library domain (study rooms) and one in the Hotel domain (hotel rooms). Figures 7 and 11 report results for their respective domains, and the numbers are consistent. The apparent discrepancy likely comes from mixing the two domains when comparing the figures.
>
> > Re C3: "Limited error analysis. While the paper shows the model performance breakdown by evaluation criteria in Figure 8, and one example erroneous trajectory in the Appendix, it is still unclear why leading models fail."
>
> Task difficulty in SOPBench is multifaceted, as each task requires the model to both *remember* to follow the SOP and *correctly* execute it. Concretely, models must demonstrate three core capabilities:
>  - **C1: Complete constraint verification**: remember to check *all* relevant constraints.
>  - **C2: Correct constraint reasoning**: select the appropriate helper function for each constraint and correctly interpret its output to determine whether the constraint is satisfied, and finally determine whether the target service action is permissible.
>  - **C3: Safe and compliant action execution**: execute only permissible user requests and refuse impermissible ones, balancing helpfulness with safety.
>
> To analyze where models fail, we categorize errors into five types, each aligned with the capabilities above:
>  - **Completely overlook constraints**: the model ignores all constraints and directly calls the target service function (C1).
>  - **Partially overlook constraints**: the model attempts constraint verification but fails to check all required constraints (C1).
>  - **Wrong interpretation**: the model calls the correct helper functions but misinterprets their outputs, leading to an incorrect final decision (C2).
>  - **Fail to complete a permissible task**: the model should execute the user request but fails to do so (C3).
>  - **Complete an impermissible task**: the model executes the target service function even though constraints prohibit it (C3).
>
> Below is the error distribution across representative models:
>
> | Model | o4-mini | gemini-2.5-flash | gpt-4.1 | claude-3.7-sonnet | llama3.1-70b-instruct | qwen2.5-32b-instruct | qwen2.5-14b-instruct |
> | :---- | :---- | :---- | :---- | :---- | :---- | :---- | :---- |
> | Completely overlook constraints | 50 | 52 | 8 | 30 | 131 | 122 | 95 |
> | Overlook partial constraints | 22 | 51 | 47 | 81 | 112 | 126 | 161 |
> | Wrong interpretation | 14 | 22 | 23 | 26 | 31 | 20 | 20 |
> | Fail to complete permissible task | 87 | 85 | 68 | 65 | 88 | 76 | 66 |
> | Complete impermissible task | 52 | 90 | 139 | 205 | 219 | 226 | 276 |
>
> [1/2]

---

> ### Author Response · Authors · 2025-11-28
> **Response to reviewer h1ei [2/2]**
>
> These models show different error patterns:
>  - **o4-mini:** Often either checks all constraints or ignores them entirely. Rarely misinterprets results, but is **overly conservative**, failing many permissible tasks and thus reducing helpfulness.
>  - **Gemini-2.5-Flash:** Misses more constraints than o4-mini and shows a **balanced mix** of failing permissible tasks and completing impermissible ones, indicating unstable safety–helpfulness behavior.
>  - **GPT-4.1 vs. Claude-3.7-Sonnet:** GPT-4.1 almost never overlooks all constraints, though it still misses partial checks. Claude-3.7 shows **more overlook errors** overall, reflecting weaker constraint-following ability. Both models more frequently **complete impermissible tasks**, suggesting safety lapses after partial reasoning.
>  - **Open-source models (Llama3.1-70B, Qwen2.5-32B/14B):** Frequently overlook constraints (partially or fully) and have the **highest impermissible completion rates**, indicating weak structural understanding of constraints and poor safety compliance.
>
> Overall, stronger models tend to remember to check constraints but may misinterpret results or act unsafely afterward. Weaker models often miss constraint checks entirely. These patterns highlight that SOPBench exposes *distinct procedural-reasoning weaknesses* across models, not just aggregate accuracy differences, and pinpoints where future model improvements are needed. We have included this analysis in the updated draft.
>
> > Re C4: "Also, what makes domains like Hotel more challenging?"
>
> The Hotel domain is designed to be challenging with some complex temporal constraints. Quantitatively, Hotel contains 10 temporal constraints, the highest of all domains. These constraints involve a uniquely diverse set of temporal operations: lead-time windows defined by two offsets, hour-level deadlines, intra-day service periods, interval-length calculations, and interval-overlap detection across existing bookings.
>
> To see this more concretely, consider the `modify_reservation` action, which is governed by four separate temporal constraints that must all hold at the same time:
>  - Within the lead-time window. The new check-in date must fall within an allowed window relative to “today.” In other words, the guest cannot move the reservation to a date that is too soon or too far in the future.
>  - No overlap with exclusion. When modifying a reservation, the assistant must ignore the original booking interval and check whether the guest has some other booking whose date range overlaps with the proposed new check-in and check-out dates.
>  - Before an hour-level deadline. The modification request must be made sufficiently in advance of arrival, measured in hours rather than days.
>  - Within the maximum stay length. The modified reservation must not exceed the maximum allowed number of nights.
>
> We found that even the leading models struggle to correctly verify these constraints in the hotel domains.
>
> > Re C5: How is evaluation done with temp = 0 when models like GPT-5 do not support temperature?
>
> For the models that does not support temperature and top\_p, we use the default inference mode exposed by the API, without specifying temperature and top\_p. We will clarify this in the paper.
>
> > Re C6: Since the constraints can be translated into a code program, how is the performance if the LLM is asked to generate code for constraint verification?
>
> The code program for checking constraints is tightly coupled with several design components, such as the database schema, service function to constraint mappings, and constraint to helper function mappings. As shown in Appendix B (Figure 12), these components were designed iteratively, and implementing the verification code is straightforward once the designs are finalized. Because this process depends on coordinated design rather than isolated code generation, we wrote the verification code manually. However, after the designs are finalized, generating code with an LLM should be straightforward.

---

### Official Review · Reviewer_6wKw · 2025-11-01

**Soundness:** 3
**Presentation:** 3
**Contribution:** 3
**Rating:** 8
**Confidence:** 3

**Summary:**

Since the research about the current language agent system perform on complying standard operating procedures (SOPs), policies, and constraints is limited, this paper aims to address the underexplored problem and introduces an automated evaluation pipeline, SOPBench. This pipeline measures procedural compliance across multi-step tool-use tasks. It constructs a benchmark consists of sandboxed environments, automated test generation, and evaluation mechanism. The authors conduct extensive experiments to evaluate the benchmarks with leading models and provide valuable insights. The release of this benchmark can benefit future research.

**Strengths:**

- The benchmark is large-scale and diverse, covering 167 executable tool functions, and 830 test cases from 7 customer-service domains. The authors also evaluate the benchmark with 18 leading models.

- The paper highlights that even current leading models struggle with some tasks and domains, which indicates that SOP control remains challenging, and the benchmark is both relevant and impactful.

- The paper also shows the vulnerabilities of current models to malicious jailbreaking, and found that easy jailbreaking can success on overlooking SOPs and constraints, which inspire safety studies using this benchmark.

- The work formalizes SOPs as directed action graphs and proposes a three-level evaluation methodology, which can provide multiple perspectives for evaluation to following studies; and the code-based verifiers provide the foundation for future work in RLVR.

- The extensive experiments regarding different metrics on different domains, models, adversarial attacks, tool use methods, and service tasks, further demonstrates the diversity of the benchmark, which can benefit various research in the community.

- The paper is well-written and well-organized, with intuitive figures and helpful examples that make it straightforward to understand.

**Weaknesses:**

- The domains can be more diverse to include some industry domains with more complex and messier SOPs and constraints, i.e., requiring larger amounts of service/helper functions and constraints.

- It could be better to categorize these tasks into different levels of difficulty.

- It may be beneficial to also show the statistics of average or deepest constraint dependency across domains, and provide additional analysis regarding how the constraint complexity impacts.

**Questions:**

- Can the current pipeline be scaled to some real scenarios with a large amount of rules and complex dependency between functions?

- The test cases are structured based on the tools and constraints, but how were the free-text user inputs generated? In addition, can the user inputs include “outlier” intents that involve constraints outside the predefined tool and constraint space?

---

> ### Author Response · Authors · 2025-11-28
> **Response to reviewer 6wKw**
>
> We thank the reviewer for the thoughtful and constructive feedback. We appreciate the recognition of our work’s strengths, including the scale and diversity of the benchmark (167 tools, 7 domains), the comprehensive evaluation with 18 models, the relevance of SOP control as a still-unsolved challenge, and the benchmark’s utility for both safety and RLVR research. We are also glad that the reviewer found the paper well-written and easy to follow.
>
> Addressing Concerns:
>
> > Re C1: "The domains can be more diverse to include some industry domains with more complex and messier SOPs and constraints, i.e., requiring larger amounts of service/helper functions and constraints.""
>
> We agree that incorporating more complex, “messy” industrial SOPs would be valuable. Importantly, SOPBench is designed as an extendable framework: adding a new domain only requires specifying the constraint graph, service/helper functions, and database schema. Our pipeline then automatically generates the corresponding test cases (see Section 2.4, Figure 4, and details in Appendix B).
>
> This structure makes it straightforward to introduce richer domains with larger toolsets, deeper dependency chains, and more intricate constraints. At the same time, our current results show that even frontier models struggle with procedural and constraint adherence on the current data, suggesting that isolating this core capability is an essential first step before scaling to heavier industrial scenarios.
>
> > Re C2: "It could be better to categorize these tasks into different levels of difficulty. It may be beneficial to also show the statistics of average or deepest constraint dependency across domains, and provide additional analysis regarding how the constraint complexity impacts."
>
> Task complexity in SOPBench is multifaceted, as each task requires the model to (1) C1: remember to check all relevant constraints in the correct order, (2) C2: select the appropriate helper function for each constraint and correctly interpret its output, and (3) C3: correctly determine whether the target service action is permissible.
>
> Figure 9 categorizes tasks by the number of constraints, which may partly reflect underlying task complexity, with corresponding performance breakdowns. As shown, model accuracy generally decreases with higher constraint complexity. The strongest reasoning models (o4-mini-high, GPT-5, and DeepSeek-R1) remain relatively stable even as constraints increase.
>
> Below are the requested statistics on **constraint dependency depth** for each domain. We add more details (depth for each task) in the Appendix in the updated draft.
>
> | Domain        | Min | Max | Median | Mean | Std  |
> |---------------|-----|-----|--------|------|------|
> | Bank          | 1   | 5   | 4      | 3.47 | 1.23 |
> | DMV           | 2   | 5   | 3      | 3.55 | 0.89 |
> | Healthcare    | 1   | 5   | 3      | 3.08 | 1.21 |
> | Online Market | 2   | 8   | 5      | 4.67 | 1.49 |
> | University    | 2   | 6   | 5      | 4.67 | 1.37 |
> | Hotel         | 1   | 4   | 3      | 2.79 | 0.94 |
> | Library       | 1   | 8   | 4      | 4.35 | 1.94 |
>
> Note that task complexity is not solely determined by the number of constraints and dependency-graph depth; tasks may be challenging for additional structural or semantic reasons.
>
> To further analyze where models fail, we categorize errors into five types, each aligned with the capabilities above:
>  - **Completely overlook constraints**: the model ignores all constraints and directly calls the target service function (C1).
>  - **Partially overlook constraints**: the model attempts constraint verification but fails to check all required constraints (C1).
>  - **Wrong interpretation**: the model calls the correct helper functions but misinterprets their outputs, leading to an incorrect final decision (C2).
>  - **Fail to complete a permissible task**: the model should execute the user request but fails to do so (C3).
>  - **Complete an impermissible task**: the model executes the target service function even though constraints prohibit it (C3).
>
> Below is the error distribution across representative models:
>
> | Model | o4-mini | gemini-2.5-flash | gpt-4.1 | claude-3.7-sonnet | llama3.1-70b-instruct | qwen2.5-32b-instruct | qwen2.5-14b-instruct |
> | :---- | :---- | :---- | :---- | :---- | :---- | :---- | :---- |
> | Completely overlook constraints | 50 | 52 | 8 | 30 | 131 | 122 | 95 |
> | Overlook partial constraints | 22 | 51 | 47 | 81 | 112 | 126 | 161 |
> | Wrong interpretation | 14 | 22 | 23 | 26 | 31 | 20 | 20 |
> | Fail to complete permissible task | 87 | 85 | 68 | 65 | 88 | 76 | 66 |
> | Complete impermissible task | 52 | 90 | 139 | 205 | 219 | 226 | 276 |
>
> [1/2]

---

> > ### Author Response · Authors · 2025-11-28
> > **Response to reviewer 6wKw [1/2]**
> >
> > These models show different error patterns:
> >  - **o4-mini:** Often either checks all constraints or ignores them entirely. Rarely misinterprets results, but is **overly conservative**, failing many permissible tasks and thus reducing helpfulness.
> >  - **Gemini-2.5-Flash:** Misses more constraints than o4-mini and shows a **balanced mix** of failing permissible tasks and completing impermissible ones, indicating unstable safety–helpfulness behavior.
> >  - **GPT-4.1 vs. Claude-3.7-Sonnet:** GPT-4.1 almost never overlooks all constraints, though it still misses partial checks. Claude-3.7 shows **more overlook errors** overall, reflecting weaker constraint-following ability. Both models more frequently **complete impermissible tasks**, suggesting safety lapses after partial reasoning.
> >  - **Open-source models (Llama3.1-70B, Qwen2.5-32B/14B):** Frequently overlook constraints (partially or fully) and have the **highest impermissible completion rates**, indicating weak structural understanding of constraints and poor safety compliance.
> >
> > Overall, stronger models tend to remember to check constraints but may misinterpret results or act unsafely afterward. Weaker models often miss constraint checks entirely. These patterns highlight that SOPBench exposes *distinct procedural-reasoning weaknesses* across models, not just aggregate accuracy differences, and pinpoints where future model improvements are needed. We have included this analysis in the updated draft.
> >
> > > Re C3: "Can the current pipeline be scaled to some real scenarios with a large amount of rules and complex dependency between functions?"
> >
> > Yes. SOPBench is explicitly designed as an extendable framework; please refer to our response to C1.
> >
> > > Re C4: "The test cases are structured based on the tools and constraints, but how were the free-text user inputs generated? In addition, can the user inputs include “outlier” intents that involve constraints outside the predefined tool and constraint space?"
> >
> > As noted in Lines 299–304, each test case begins with a single free-text user request containing the target action description (the user’s goal achievable via a service function) and the user-known information required for completing the task and checking constraints. These components are first generated by an LLM in structured form (detailed in Section 2.4, illustrated in Figure 4, with additional details in Appendix B), and the final free-text request is produced by prompting the LLM to paraphrase and combine them. Because the content is fully specified beforehand, the resulting user inputs are tightly controlled and do not introduce intents outside the defined tool/constraint space. We have also manually verified the dataset and confirm that no outlier intents appear in the final benchmark.

---

### Meta-Review · Area_Chair_pj8L · 2025-12-23

**Summary:**

This paper introduces SOPBench, a benchmark evaluating language agents on adherence to standard operating procedures and constraints across seven customer service domains with 167 executable tools and 830 test cases. The work addresses procedural compliance in multi-step tool use through automated test generation and oracle-based verification at outcome, step, and trajectory levels. Strengths include the novel focus on SOP compliance rather than mere task completion, comprehensive coverage with rigorous evaluation methodology, demonstration of significant vulnerabilities even in frontier models, and the release of datasets and trajectories. However, weaknesses outweigh these contributions. The core novelty beyond existing tool-calling benchmarks remains unclear, with insufficient differentiation from function-calling leaderboards that also perform state-based evaluation. The domains lack diversity beyond customer service, potentially limiting real-world applicability to messier industrial scenarios. Error analysis is superficial, failing to explain why leading models struggle or what makes certain domains more challenging. The benchmark difficulty may exceed current capabilities without providing actionable insights for improvement. Most critically, the fundamental motivation is questionable since well-defined SOPs are better implemented directly in code rather than through LLM agents. The decision to reject stems from limited technical novelty, narrow scope, and unclear practical utility for the proposed use case.

**Reviewer Concerns:**

The authors provided detailed rebuttals addressing reviewer concerns. They clarified that SOPBench evaluates procedural safety compliance rather than tool proficiency, distinguishing it from completion-focused benchmarks. They added comprehensive error taxonomy analysis categorizing failures into constraint overlooking, wrong interpretation, and unsafe execution patterns across models. The rebuttal explained Hotel domain complexity through temporal constraint analysis and provided constraint dependency statistics. Authors emphasized the framework's extensibility for adding domains and justified LLM agent evaluation for open environments where backend control is unavailable. Reviewer 6wKw found responses satisfactory regarding extensibility and error analysis. Reviewers h1ei and 8YNX raised concerns about novelty versus tool-calling benchmarks and questioned whether LLMs are appropriate for rule-based tasks, which the rebuttal addressed but likely did not fully resolve given the fundamental conceptual disagreement. Reviewer o9er's concerns about manual implementation bias and narrow domain focus were partially addressed through extensibility arguments but the generalizability concerns remain valid. The error analysis addition strengthens the paper but does not overcome the core novelty and motivation issues.

**Reviewer Scores:**

Reviewer 6wKw would likely maintain their score of 8, as their concerns were comprehensively addressed with concrete additions. Reviewers h1ei and 8YNX would likely remain at 4, as their fundamental concerns about novelty differentiation from tool-calling benchmarks and the appropriateness of using LLMs for codifiable SOPs were acknowledged but not convincingly resolved. Reviewer o9er would likely stay at 4, since while the extensibility argument addresses scope concerns, the fundamental limitations of manual construction and narrow applicability persist. The overall consensus suggests marginally below threshold with three reviewers at 4 and one at 8, indicating the work has merit but insufficient novelty and unclear positioning relative to existing benchmarks.

---

### Decision · Program_Chairs · 2026-01-26

Reject